# Modeling Dengue Immune Responses Mediated by Antibodies: A Qualitative Study

**DOI:** 10.3390/biology10090941

**Published:** 2021-09-20

**Authors:** Afrina Andriani Sebayang, Hilda Fahlena, Vizda Anam, Damián Knopoff, Nico Stollenwerk, Maíra Aguiar, Edy Soewono

**Affiliations:** 1Department of Mathematics, Institut Teknologi Bandung, Bandung 40132, Indonesia; afrina.andriani@s.itb.ac.id (A.A.S.); hildafahlena@s.itb.ac.id (H.F.); 2Basque Centre for Applied Mathematics (BCAM), Alameda Mazarredo, 14, 48009 Bilbao, Spain; vanam@bcamath.org (V.A.); dknopoff@bcamath.org (D.K.); nstollenwerk@bcamath.org (N.S.); 3Dipartimento di Matematica, Universita degli Studi di Trento, Via Sommarive 14, 38123 Trento, Italy; 4Ikerbasque, Basque Foundation for Science, Euskadi Plaza, 5, 48009 Bilbo, Spain; 5Center for Mathematical Modeling and Simulation, Institut Teknologi Bandung, Bandung 40132, Indonesia

**Keywords:** within-host modeling, dengue fever, immune response, antibodies, viral load, antibody-dependent enhancement

## Abstract

**Simple Summary:**

With more than one-third of the world population at risk of acquiring the disease, dengue fever is a major public health problem. Caused by four antigenically distinct but related serotypes, disease severity is associated with the immunological status of the individual, seronegative or seropositive, prior to a natural dengue infection. While a primary natural dengue infection is often asymptomatic or mild, individuals experiencing a secondary dengue infection with a heterologous serotype have higher risk of developing the severe form of the disease, linked to the antibody-dependent enhancement (ADE) process. We develop a modeling framework to describe the dengue immune responses mediated by antibodies. Our model framework can describe qualitatively the dynamic of the viral load and antibodies production for scenarios of primary and secondary infections, as found in the empirical immunology literature. Studies such as the one described here serve as a baseline to further model extensions. Future refinements of our framework will be of use to evaluate the impact of imperfect dengue vaccines.

**Abstract:**

Dengue fever is a viral mosquito-borne infection and a major international public health concern. With 2.5 billion people at risk of acquiring the infection around the world, disease severity is influenced by the immunological status of the individual, seronegative or seropositive, prior to natural infection. Caused by four antigenically related but distinct serotypes, DENV-1 to DENV-4, infection by one serotype confers life-long immunity to that serotype and a period of temporary cross-immunity (TCI) to other serotypes. The clinical response on exposure to a second serotype is complex with the so-called antibody-dependent enhancement (ADE) process, a disease augmentation phenomenon when pre-existing antibodies to previous dengue infection do not neutralize but rather enhance the new infection, used to explain the etiology of severe disease. In this paper, we present a minimalistic mathematical model framework developed to describe qualitatively the dengue immunological response mediated by antibodies. Three models are analyzed and compared: (i) primary dengue infection, (ii) secondary dengue infection with the same (homologous) dengue virus and (iii) secondary dengue infection with a different (heterologous) dengue virus. We explore the features of viral replication, antibody production and infection clearance over time. The model is developed based on body cells and free virus interactions resulting in infected cells activating antibody production. Our mathematical results are qualitatively similar to the ones described in the empiric immunology literature, providing insights into the immunopathogenesis of severe disease. Results presented here are of use for future research directions to evaluate the impact of dengue vaccines.

## 1. Introduction

Dengue fever is a viral mosquito-borne infection affecting a large percentage of the population living in the tropics and subtropics. Caused by four antigenically related but distinct viruses, DENV-1, DENV-2, DENV-3 and DENV-4, it is estimated that around 400 million dengue infections occur every year [1], with disease severity being influenced by the immunological status of the individual, seronegative or seropositive, prior to natural infection. While a primary dengue infection is usually asymptomatic or results in mild disease manifestation, the immunological response on exposure to a heterologous dengue serotype is complex, recognized to be a risk factor of progressing to severe disease [2,3,4,5,6,7].

Early dengue diagnosis is important for the clinical management of the patient [8,9]. The most commonly used technique for dengue routine diagnosis is the enzyme-linked immunosorbent assay (ELISA), with primary or secondary infections being characterized based on the concentration of immunoglobulins M and G from the blood sample, the so-called IgM and IgG antibodies, respectively [10,11,12].

From the basic immunology literature, it is known that the IgM is the first antibody secreted by the adaptive immune system in response to a foreign antigen, followed by the production of IgG antibodies with increased affinity for the pathogen causing the infection [13,14]. Likewise, in a primary dengue infection, the IgM antibody type is produced more quickly and to higher levels than the IgG antibody type, and the reverse is true in secondary dengue infection [6,7]. In addition to conferring life-long protective immunity against a specific serotype, the IgG antibody is able to cross-react with heterologous DENV-serotypes [6,7,15,16,17]. Instead of neutralizing the new dengue serotype, the pre-existing antibodies promote the enhancement of the infection by facilitating the entry of the complex antibody-heterologous virus into target cells. This disease augmentation phenomenon is called antibody-dependent enhancement (ADE) [3,6,7,18,19] and its occurrence in dengue has been used to explain the etiology of severe disease [7,20,21,22], which has been shown to be correlated with higher viral loads [23,24,25,26].

Treatment of uncomplicated dengue cases is only supportive, and severe dengue cases require careful attention to fluid management and proactive treatment of hemorrhagic symptoms. Two tetravalent dengue vaccines have completed phase 3 clinical trial: Dengvaxia, a product developed by Sanofi Pasteur that is now licensed in several countries [27], and the DenVax vaccine, developed by Takeda Pharmaceutical Company [28,29]. While Dengvaxia has resulted in serious adverse events in seronegative individuals compared with age-matched seronegative controls [30,31,32,33], long-term surveillance consisting of prudent and careful observation of DenVax vaccine recipients is required, since negative vaccine efficacy was estimated for vaccinated seronegative individuals who were infected with serotype 3, as opposed to an intermediate efficacy for seropositive [34,35].

In recent years, mathematical modeling became an important tool for the understanding of infectious disease epidemiology and dynamics, at both macroscopic and microscopic levels, addressing ideas about the components of host-pathogen interactions. Dengue models are often used to understand infectious disease dynamics and to evaluate the introduction of intervention strategies such as vector control and vaccination. At the population level, multi-strain dengue dynamics have been modeled with extended (susceptible-infected-recovered) SIR-type models including immunological aspects of the disease such as temporary cross-immunity and ADE phenomenology [36,37,38,39,40,41,42]. However, within-host host dengue modeling is restricted to a small number of studies so far [43,44,45,46,47,48]. Within-host models consider the dynamic interaction between free virus and susceptible target cells [43,44,45], differing on the functional form used to model viral infectivity, immune response, and viral clearance dynamics. However, the role of pre-existing DENV-serotype specific IgG antibody in a secondary dengue infection with an explicit mechanism to explain its protective or enhancing effect has not been deeply explored yet.

In this paper, we present a mathematical model framework developed to describe the dengue immunological response mediated by antibodies. Three models are analyzed and compared: (i) primary dengue infection, (ii) secondary dengue infection with the same (homologous) serotype and (iii) secondary dengue infection with a heterologous dengue virus. The model is a refined version to that proposed in [43], and can describe qualitatively the dynamics of viral load and antibody production and decay for scenarios of primary and secondary infections as found in the empirical immunology literature [3,6,7,15,18,19,22]. Providing insights into the immunopathogenesis of severe diseases, the results presented here are of use for future research directions to evaluate the impact of dengue vaccines.

## 2. Modeling Within-Host Dengue Infections

In the absence of good laboratory data, the aim of this study is to describe qualitatively the dynamics of viral load and antibodies responses during dengue infections. We also evaluate the effects of pre-existing antibodies, produced during a primary dengue infection, on a secondary dengue infection with the same serotype (homologous serotype), and secondary dengue infection with a different serotype (heterologous serotype).

In this section, we present the models developed to describe dengue immunological responses mediated by antibodies. A minimalistic mathematical modeling framework considering primary and secondary dengue infections is proposed, with models developed by adding gradually the steps of disease infection and immunological responses, as described in the immunology literature. The proposed models are based on body cells and free viral particle interactions that result in infected cells and subsequently trigger the activation of the immune response mediated by antibodies. We explore the feature of viral replication, viral load, antibody production, antibody activation and antibody decay, as well as the infection clearance process during a primary dengue infection, a secondary dengue infection with homologous serotype and a secondary dengue infection with a heterologous serotype, where the process of ADE is expected to occur.

### 2.1. Primary Dengue Infection Model

Dengue viruses are transmitted to a human host by an infected female *Aedes* mosquito bite. It is called a primary dengue infection if it occurs in seronegative hosts, i.e., individuals with no history of previous dengue infections. In its simplicity, the interaction between target cells, infected cell, virus, and immunological response mediated by antibodies is represented in Figure 1.

Briefly, susceptible target cells, monocytes, and dendritic cells (*S*) are produced by the body at a constant rate ($\pi_{S}$) and have a natural mortality rate $\mu_{S}$, where $\frac{1}{\mu_{S}}$ is the expected lifetime of the uninfected, i.e., susceptible target cell. Free dengue virus *V* infects susceptible target cells *S* at rate *a*, producing infected cells *I* (see process 1 in Figure 1) [49]. It is assumed that infected cells have an infection-induced mortality rate $\mu_{i}\geq \mu_{S}$, releasing free virus κ to the system (see process 2 in Figure 1). We assume that several free virus particles are needed to infect a single susceptible cell and therefore, while the number of susceptible cells decreases with aSV rate, the number of free viruses decreases with a bSV rate.

Macrophages are also considered in the system as a target susceptible cell $S_{m}$. Upon infection, those cells differentiate to become presenting cells (*P*), shown in process 3 in Figure 1. Presenting cells are assumed to trigger, via antigen presentation, the production of antibodies IgM (*M*) and IgG (*G*) with rates $\alpha_{M}$ and $\alpha_{G}$, respectively (see process 4 and 5 in Figure 1. Presenting cells can eventually die with antigen presentation induced mortality rate $\mu_{P}$.

While in a primary infection IgM antibodies are produced first and to higher levels than IgG, the reverse is true in a secondary infection. IgM antibody (a pentamer molecule) and IgG antibody (a monomer molecule) [50], bind into the free virus with rates $\gamma_{M}d_{M}$ and $\gamma_{G}d_{G}$, generating antibody-virus complexes IgM-DENV ($C_{M}$) and IgG-DENV ($C_{G}$), respectively (see process 6 and 7 in Figure 1) [18,51]. Those complexes are assumed to clear the ongoing infection after being recognized by killing cells.

In order to understand the individual dynamics of viral replication, viral load, antibodies production and decay, and finally the clearance of infection, our model is constructed in blocks of equations which are coupled gradually until we obtain the complete model framework able to describe a primary dengue infection and its immunological response mediated by antibodies.

#### 2.1.1. Virus Replication Dynamics

With susceptible target cells (monocytes and dendritic cells) *S*, infected cells *I*, and the virus *V*, the process of viral replication can be analyzed with a basic SIV model as follows
(1)$$\begin{array}{ccc} \frac{dS}{dt} & = & \pi_{S}-\mu_{S}S-aSV\\ \frac{dI}{dt} & = & aSV-(\mu_{i}+\mu_{S})I\\ \frac{dV}{dt} & = & \kappa \mu_{i}I-bSV, \end{array}$$
where all parameters are described in Table 1.

The model described in Equation System (Equation 1) shows an exponential growth of viral particles in the absence of any immunological response. The free viral growth depends on the virus replication factor κ, as well as by the infection rate of susceptible cells *a* and the removal rate of viral particles during the infection of susceptible cells *b*. As the values of parameters are shown in Table 1, the numerical simulations are shown in Figure 2, with free virus detected around day 2 of the infection process.

To investigate the sensitivity of viral level related to the model parameters in Equation System (Equation 1), Figure 3 presents the numerical result of viral load related to the viral replication factor κ, the infection rate of susceptible target cells *a*, and the removal rate of viral particles *b*. Sensitivity analysis is performed by varying one of the parameters and fixing the others. The result shows that the variation of the number of free viral particles released by an infected cell plays a major role in viral load peak, reaching very high values in a short period of time as κ increases (see Figure 3a).

As for the infection rate of susceptible cells *a*, free viral particle levels increase as the parameter value increases, since a higher infection rate generates more infected cells that will release more viral particles. The biological time for free viral particles detection also decreases as parameter *a* increases, as shown in Figure 3b. On the other hand, only a small variation of free viral load particles is observed when changing the rate *b*, at which the viral particles are lost due to the infection process, as shown in Figure 3c.

#### 2.1.2. IgM and IgG Antibody Production and Decay and Free Viral Load Dynamics

To understand the process of antibody production via antigen presentation, we now extend the Equation System (3) to include another susceptible target cell type, the macrophages ($S_{m}$). Upon infection, macrophages will differentiate to become antigen-presenting cells *P*, triggering the production of free IgM and free IgG antibodies types [13,33,54] at rates αM and αG, respectively. In a primary infection, IgM antibodies, a pentamer molecule, are produced first and to higher levels than IgG antibodies, a monomer molecule [50]. Free IgM and free IgG bind into the free viral particles with $d_{M}\gamma_{M}$ and $d_{G}\gamma_{G}$ binding rates, respectively.

The extended model to describe the IgM and IgG production is given by
(2)$$\begin{array}{ccc} \frac{dS}{dt} & = & \pi_{S}-\mu_{S}S-aSV\\ \frac{dI}{dt} & = & aSV-(\mu_{i}+\mu_{S})I\\ \frac{dV}{dt} & = & \kappa \mu I-bSV-b_{m}S_{m}V-d_{M}MV-d_{G}VG\\ \frac{S_{m}}{dt} & = & \pi_{m}-\mu_{S}S_{m}-a_{m}S_{m}V\\ \frac{dP}{dt} & = & a_{m}S_{m}V-\left(\mu_{P}+\mu_{S}\right)P\\ \frac{dM}{dt} & = & \alpha_{M}P-\gamma_{M}MV-\mu_{M}M\\ \frac{dG}{dt} & = & \alpha_{G}P-\gamma_{G}GV-\mu_{G}G, \end{array}$$
including natural removal for IgM, $\mu_{M}M$, and IgG, $\mu_{G}G$.

Free IgM antibody production is observed to start between day 2 and day 3 of the infection process (see Figure 4a), lasting for about three months (see Figure 4b). Free IgG antibody type appears shortly after IgM antibodies (see Figure 4a), with lower concentration levels, but lasting much longer than free IgM (see Figure 4b), reaching eventually a constant “life-long immunity” level. Viral load dynamics (see Figure 4c) is influenced by the antibodies production, with a peak between day 5 and day 6 of the infection process. The complete process of free virus dynamics in the presence of antibodies is shown in Figure 4d.

#### 2.1.3. Antibody-Virus Complexes and Infection Clearance

Following the antibody production process described above, the model framework is extended to include the antibody-virus complex production, IgM-DENV ($C_{M}$) and IgG-DENV ($C_{G}$), which are assumed to be responsible for clearing the ongoing infection after being recognized by killing cells. With constant target cells production $\pi_{S}$, for monocytes and dendritic cells, and $\pi_{m}$ for macrophages, the complete modeling framework including each step presented in Figure 1 is written as a system of ordinary differential equations (ODEs) as follows
(3)$$\begin{array}{ccc} \frac{dS}{dt} & = & \pi_{S}-aSV-\mu_{S}S\\ \frac{dI}{dt} & = & aSV-(\mu_{i}+\mu_{S})I\\ \frac{dV}{dt} & = & \kappa \mu_{i}I-bSV-b_{m}S_{m}V-d_{M}VM-d_{G}VG\\ \frac{dS_{m}}{dt} & = & \pi_{m}-a_{m}S_{m}V-\mu_{S}S_{m}\\ \frac{dP}{dt} & = & a_{m}S_{m}V-\left(\mu_{P}+\mu_{S}\right)P\\ \frac{dM}{dt} & = & \alpha_{M}P-\gamma_{M}MV-\mu_{M}M\\ \frac{dG}{dt} & = & \alpha_{G}P-\mu_{G}G-\gamma_{G}VG\\ \frac{dC_{M}}{dt} & = & \gamma_{M}VM-\mu_{C_{M}}C_{M}\\ \frac{dC_{G}}{dt} & = & \gamma_{G}VG-\mu_{C_{G}}C_{G}. \end{array}$$

The complete model output describing the immunological response mediated by IgM and IgG antibodies during a primary dengue infection is shown in Figure 5.

In Figure 5, the overall viral load curve (in red) includes not only free viral particles, as shown in Figure 4c, but also the viral particles bound into antibody-virus complexes. Free IgM (in violet) are observed at very low levels until day 5 of infection since the majority of the molecules are bound to the free virus, the so-called IgM-DENV complexes (in blue). Note that for each IgM, four viral particles must be counted on average. Free IgM appears to be detectable on day 9 after the infection is cleared, i.e., removal of all $C_{M}$ complexes, lasting for about three months. Free IgG (in green) and IgG-DENV complex (in orange) are appearing around day 4, and eventually do not play a significant role in the primary infection clearance. Free IgG reaches very small levels in comparison with the free IgM, lasting much longer than IgM, and are assumed to confer lifelong immunity against that specific serotype.

### 2.2. Secondary Dengue Infection Model with a Homologous Serotype

After a period of temporary cross-immunity, the human host is considered to be susceptible again, able to acquire a secondary dengue infection [36]. In this section, we investigate a secondary infection with the same (homologous) serotype, represented in Figure 6. The difference here lies in the order of the detection of the antibody levels, IgM and IgG, in comparison with the dynamics described for the primary dengue infection, see step 4a and step 4b in Figure 6. Here, the immunological response initiates with the presence of free virus activating the pre-existing IgG antibody at rate $\alpha G_{sec}V$, shown in step 4a in Figure 6. The antibody activation process occurs faster than the adaptive humoral response, with antibody production triggered by antigen presentation, see step 5, referring to the production of the IgM antibody type, and step 4b, referring to the production of the IgG antibody type [6,7,13,14]. With that, the overall IgG levels are reaching much higher levels than the levels observed in a primary infection.

We use the same modeling framework described in Equation System (3), only including an extra term $\alpha_{G_{,sec}}V$ (shown in blue), representing the activation of the pre-existing IgG antibodies that were produced during the primary dengue infection. The complete model for the secondary dengue infection with a homologous serotype can be written as follows
(4)$$\begin{array}{ccc} \frac{dS}{dt} & = & \pi_{S}-aSV-\mu_{S}S\\ \frac{dI}{dt} & = & aSV-(\mu_{i}+\mu_{S})I\\ \frac{dV}{dt} & = & \kappa \mu_{i}I-bSV-b_{m}S_{m}V-d_{M}VM-d_{G}VG\\ \frac{dS_{m}}{dt} & = & \pi_{m}-a_{m}S_{m}V-\mu_{S}S_{m}\\ \frac{dP}{dt} & = & a_{m}S_{m}V-\left(\mu_{P}+\mu_{S}\right)P\\ \frac{dM}{dt} & = & \alpha_{M}P-\gamma_{M}MV-\mu_{M}M\\ \frac{dG}{dt} & = & \alpha_{G}P-\mu_{G}G-\gamma_{G}VG+\alpha_{G_{sec}}V\\ \frac{dC_{M}}{dt} & = & \gamma_{M}VM-\mu_{C_{M}}C_{M}\\ \frac{dC_{G}}{dt} & = & \gamma_{G}VG-\mu_{C_{G}}C_{G}, \end{array}$$
now with the immunological response initiated by the activation of the pre-existing IgG antibodies, specific to the serotype causing the primary infection. In the present scenario, these pre-existing specific IgG antibodies are able to bind and neutralize the homologous dengue serotype causing the secondary infection.

Figure 7 shows a numerical simulation of the model for the dengue immunological response during a secondary infection with a homologous serotype. The activation rate for the pre-existing IgG antibody is set to $\alpha_{G_{,sec}}=$ 2000. The faster increasing pre-existing IgG is responsible for neutralizing the free viral particles, leading to a lower overall viral load (in red). The immunological response mediated by antibodies is reversed to the response described for the primary infection, with high levels of IgG appearing before the IgM.

In this scenario, the complex IgG-DENV (in orange) plays a major role during the viral clearance (see step 6 in Figure 6 due to its specificity, being able to quickly bind and neutralize the homologous virus. Here, our model’s results show the complexes CG (in orange) appearing already on day 2 of the infection process, binding into the free viral particles, with an important role during the clearance of the ongoing infection.

### 2.3. Secondary Infection with a Heterologous Serotype

Similar to the process described in Section 2.2, we now investigate the dynamics of a secondary infection caused by a heterologous dengue serotype, recognized to be a risk factor of progressing to severe disease. The difference here lies in the ability of pre-existing IgG antibodies to bind into the new viral particles (see step 6 in Figure 8) and enhance viral replication due to the antibody-dependent enhancement (ADE) phenomenon (see step 7 in Figure 8) since these pre-existing IgG antibodies are not able to neutralize the new virus.

We use the same modeling framework described in Equation System (4), now including extra terms $a_{ADE}SC_{G}$ and $b_{ADE}SC_{G}$ (shown in violet) affecting the viral replication of the system, with an enhancement mediated by the complexes of pre-existing IgG-DENV. The complete model for the secondary dengue infection with a heterologous dengue serotype can be written as follows
(5)$$\begin{array}{ccc} \frac{dS}{dt} & = & \pi_{S}-aSV-\mu_{S}S-a_{ADE}SC_{G}\\ \frac{dI}{dt} & = & aSV-\left(\mu_{i}+\mu_{S}\right)I+a_{ADE}SC_{G}\\ \frac{dV}{dt} & = & \kappa \mu_{i}I-bSV-b_{m}S_{m}V-d_{M}VM-d_{G}VG\\ \frac{dS_{m}}{dt} & = & \pi_{m}-a_{m}S_{m}V-\mu_{S}S_{m}\\ \frac{dP}{dt} & = & a_{m}S_{m}V-\left(\mu_{P}+\mu_{S}\right)P\\ \frac{dM}{dt} & = & \alpha_{M}P-\gamma_{M}MV-\mu_{M}M\\ \frac{dG}{dt} & = & \alpha_{G}P-\mu_{G}G-\gamma_{G}VG+\alpha_{G_{sec}}V\\ \frac{dC_{M}}{dt} & = & \gamma_{M}VM-\mu_{C_{M}}C_{M}\\ \frac{dC_{G}}{dt} & = & \gamma_{G}VG-\mu_{C_{G}}C_{G}-b_{ADE}SC_{G}. \end{array}$$

Figure 9 shows the simulation for the immune response during a secondary infection with a heterologous dengue serotype. With a much higher overall viral load (in red), the immunological response mediated by antibodies is similar to the described secondary response with the same virus. However, in this scenario, the pre-existing IgG-DENV complexes (in orange) play a major role in viral replication enhancement (see step 7 in Figure 8) via the ADE process. As the viral replication continues, the adaptive humoral response produces high levels of the IgM antibody type (see step 8 in Figure 8). These high levels of IgM are assumed to play a major role during the clearance of the ongoing infection, similarly to the process observed for a primary infection (see Figure 10c). However, the disease augmentation and the much higher viral load observed in this scenario (see Figure 10e) leads to more severe clinical symptoms, including hemorrhagic manifestations that without proper treatment may lead to shock and death. The production of the new specific IgG antibody type (see step 4b in Figure 8) only occurs later and at a very small concentration level.

Note that for this study, we focus on the qualitative behavior of the dengue immunological responses. Concentrations of viral particles and antibodies are given as arbitrary but reasonable values. Model parameters are shown in Table 1, including the biological meaning and values used for the numerical simulations. We use the same initial conditions to perform the simulation of all scenarios: primary infection, secondary infection with homologous serotype and secondary infection with heterologous serotype. All the initial conditions are stated in Table 1, except the ones starting with a value of zero.

## 3. Antibody Responses and Viral Load Levels to Explain Disease Symptoms and Severity

In our within-host modeling approach, we show different dengue immunological responses mediated by antibodies. For each infection process, the IgM-antibody and IgG-antibody dynamics are shown in Figure 10.

In a primary dengue infection, the antibody IgM type is the dominant antibody type. IgM binds into the free virus and generates the antibody-virus complexes in the early stage of the infection (see Figure 10a), reaching high levels and decaying after 3 months approximately (see Figure 10b) [6,13,14]. The specific antibody IgG is produced afterward and will provide the so-called long-life specific immunity. This specific antibody maintains an immunological memory and is able to bind and to neutralize a homologous dengue serotype (see Figure 10c,d). Free virus peaks around day 5–6 of the infection process, with a fast decay reaching undetectable levels after day 8 of the infection process (see Figure 10e). A primary infection is often asymptomatic and that is eventually correlated with the viral load levels generated during a primary dengue infection.

During a second infection with a homologous serotype, the pre-existing antibody IgG type is the dominant antibody type. These antibodies immediately respond to the new serotype (see Figure 10c), able to neutralize the virus, leading to a much faster clearing of the infection. These antibodies are lasting longer, boosting the immune system of the individual, assumed to confer lifelong immunity against that specific serotype (see Figure 10d). Free virus peaks around day 4–5 and reaches a much lower viral load level than in a primary infection (see Figure 10e). Here, we assume that individuals would have no symptoms at all and eventually will not be able to transmit the disease, given the observed viral load level.

In a second infection with a heterologous serotype, the pre-existing antibody IgG type immediately responds to the new serotype (see Figure 10c), reaching very high levels. These antibodies are able to bind to the heterologous dengue serotype, but instead of neutralizing the virus, it enhances the infection (see Figure 10e). This process is called antibody-dependent enhancement (ADE), well reproduced by our system, leading to a much higher viral load level than in a primary infection. Free viral load peaks a bit earlier than in a secondary infection with a homologous virus. Here, we assume that individuals would have symptoms and eventually develop the severe form of the disease, the so-called dengue hemorrhagic fever that without proper treatment will evolve to shock syndrome and eventually death.

## 4. Conclusions

We have developed a within-host dengue modeling framework to describe qualitatively the dengue infection immunological response mediated by antibodies. Models for a primary dengue infection, secondary dengue infection with the same virus and secondary dengue infection with a different dengue virus were analyzed and compared. We have explored the features of viral replication, antibody production, activation and decay, as well as the process of infection clearance over time, including the path for disease severity via the ADE process.

Models were developed by gradually adding the steps of disease infection and the adaptive immune response described in the immunology literature. The proposed equation systems were derived from the illustrative schemes, describing each of the dengue infection process steps individually, representing a primary dengue infection (Figure 1), secondary dengue infection with homologous serotype (Figure 6), and secondary dengue infection with heterologous serotype (Figure 8). The models were developed in blocks of equations, which are coupled gradually until we obtain the complete model framework. In the absence of a significant amount of laboratory data, the aim of this study is to describe qualitatively the dengue immunological responses mediated by antibodies and to explore the feature of antibody production and ADE when pre-existing antibodies are present in the human host. Studies such as the one described here serves as a baseline for further model extensions.

Our models were able to reproduce qualitatively the features of primary and secondary dengue infections, including the ADE process leading to the disease enhancement phenomenon in a secondary dengue infection caused by a heterologous serotype. The models are, in their minimalistic format, able to give insights into the unknown biological parameters to be estimated when empirical data of viral load and antibodies concentration levels are available.

The proposed modeling framework is the first one to describe qualitatively the dynamics of viral load and antibody production, activation and decay for scenarios of primary and secondary infections, as found in the empirical immunology literature. Providing insights into the immunopathogenesis of severe diseases via pre-existing antibodies and the ADE process, the results presented here are of use for future research evaluating the impact of imperfect dengue vaccines.

## Figures and Tables

**Figure 1 biology-10-00941-f001:**
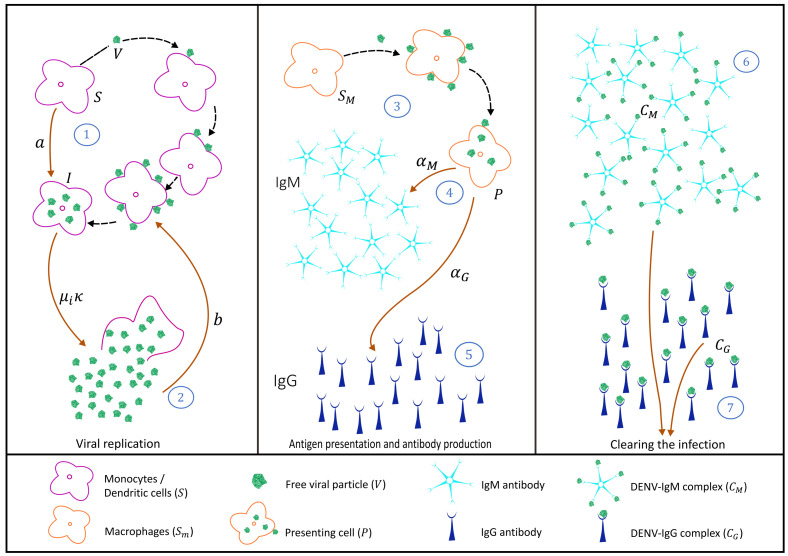
Schematic in-host dengue immunological responses mediated by antibodies: primary infection. Three blocks are used to describe 7 steps during the infection, from viral replication up to infection clearance.

**Figure 2 biology-10-00941-f002:**
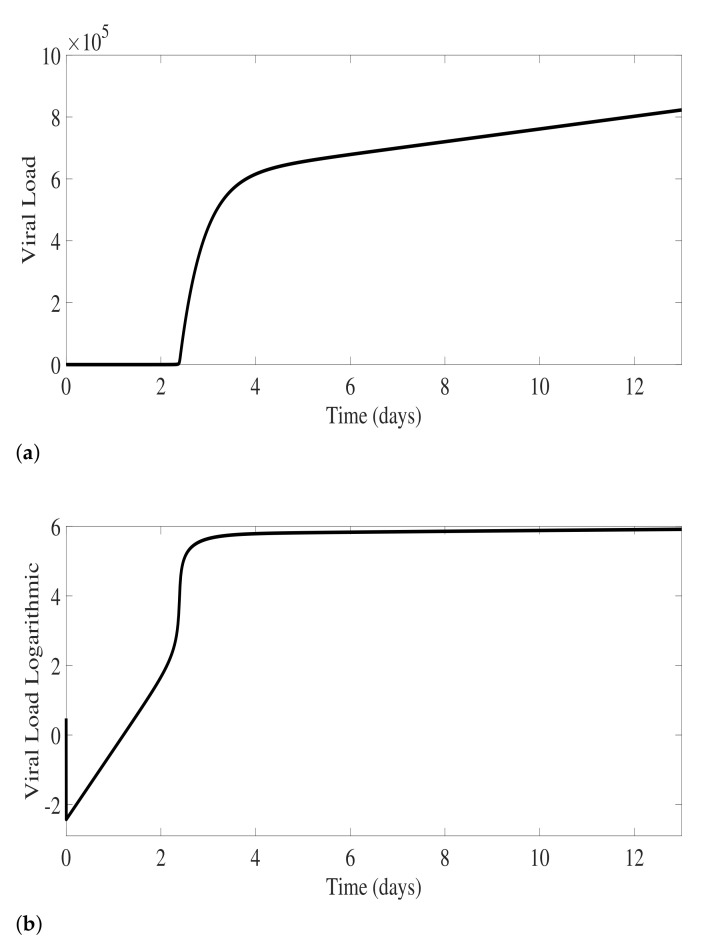
Free virus dynamics for primary infection prior to antibody production. The viral replication dynamics are shown in natural scale (**a**) and in semi-logarithmic scale (**b**) with initial value $S\left(t_{0}\right)=\pi_{S}/\mu_{S}$, $I(t_{0})=0$, and $V(t_{0})=3$. Model parameters are shown in Table 1.

**Figure 3 biology-10-00941-f003:**
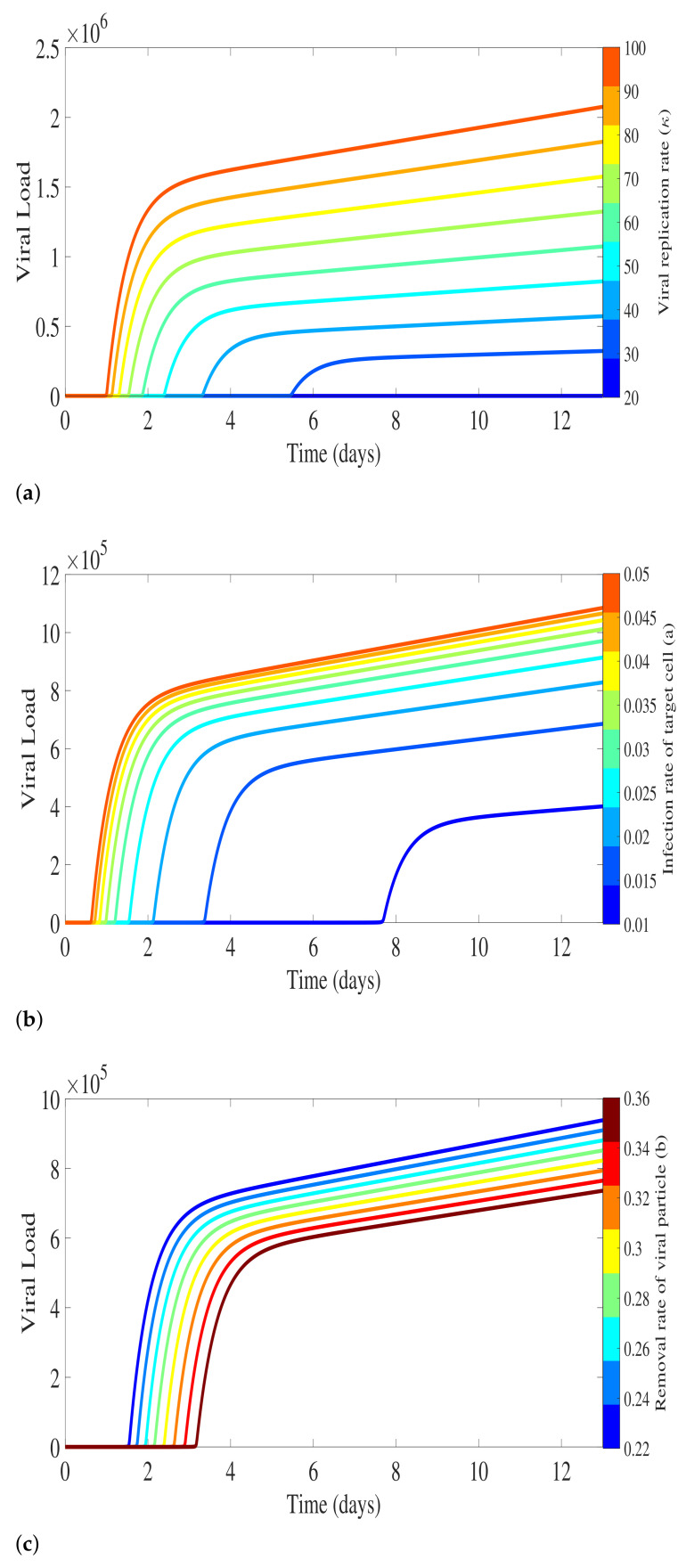
Sensitivity analysis for the parameters involved on free virus dynamics. (**a**) For fixed a=0.02 and b=15a parameters, we vary the viral replication factor κ in the range [20,100]. (**b**) For fixed κ=50 and b=0.3, we vary the infection rate of susceptible cells parameter *a* in the range [0.01,0.05]. (**c**) The removal rate of viral particles *b* is varied in the range [11a,18a] with of fixed κ=50 and a=0.02.

**Figure 4 biology-10-00941-f004:**
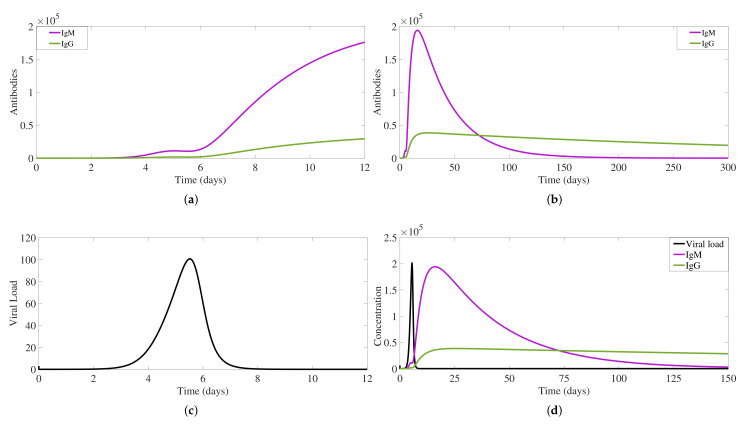
For a primary dengue infection, antibodies IgM (in violet) and IgG (in green) production dynamics are shown for a 10 days period (**a**) and for a 300 days period (**b**). Free virus particle dynamics for a 12 days period is shown in (**c**). The complete process of viral load in the presence of antibodies is shown in (**d**). Here, for better visualization, free viruses were scaled to 2000. The initial values used for these simulations are $S\left(t_{0}\right)=\pi_{S}/\mu_{S}$, $I(t_{0})=0$, $S_{m}\left(t_{0}\right)=\pi_{M}/\mu_{S}$, $P(t_{0})=0$, $M(t_{0})=0$, and $G(t_{0})=0$.

**Figure 5 biology-10-00941-f005:**
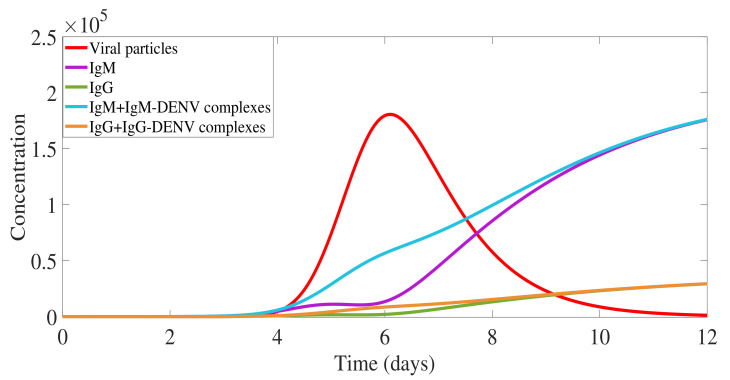
Model simulation: primary dengue infection immunological responses mediated by antibodies. Overall viral particles measured with free virus and viral particles in complexes (V+4· IgM-DENV + IgG-DENV). Free IgM is shown in violet and free IgG in green. Antibodies-virus complexes IgM-DENV and IgG-DENV are shown in blue and orange, respectively, with initial values $C_{M}\left(t_{0}\right)=0$ and $C_{G}\left(t_{0}\right)=0$.

**Figure 6 biology-10-00941-f006:**
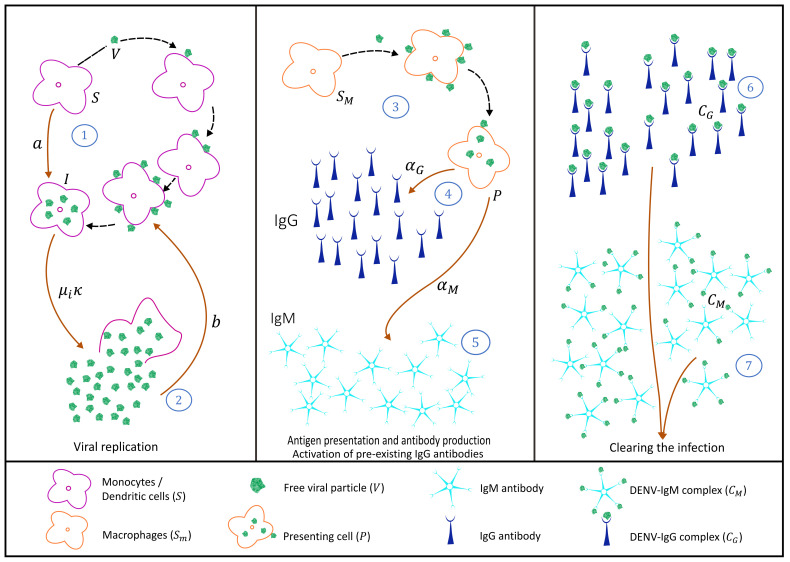
Schematic in-host dengue immunological responses mediated by antibodies: secondary infection with the same dengue serotype. Three blocks are used to describe 7 steps during the infection, from viral replication up to infection clearance.

**Figure 7 biology-10-00941-f007:**
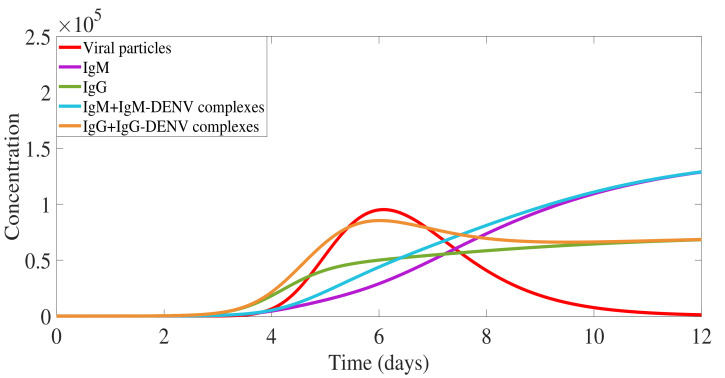
Model simulation: secondary dengue infection with a homologous serotype. Overall viral particles measured with free virus and viral particles in complexes ((V+4· IgM-DENV + IgG-DENV)). Free IgM is shown in violet and free IgG in green. Antibodies-virus complexes IgM-DENV and IgG-DENV are shown in blue and orange, respectively. The initial value of IgG antibody is set, $G(t_{0})=1000$.

**Figure 8 biology-10-00941-f008:**
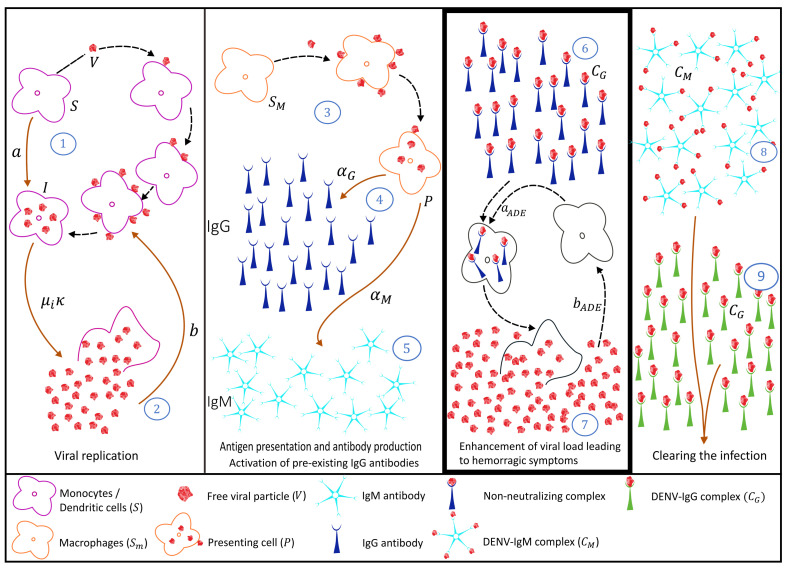
Schematic in-host dengue immunological responses mediated by antibodies: secondary infection with a different dengue serotype. Four blocks are used to describe 9 steps during the infection, from viral replication up to infection clearance, including disease augmentation via the ADE process (steps 6 and 7).

**Figure 9 biology-10-00941-f009:**
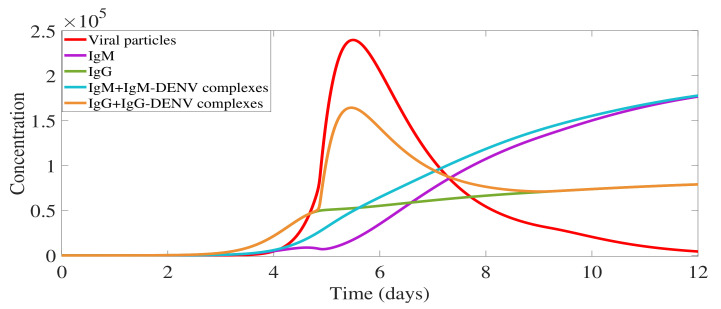
Model simulation: secondary dengue infection with a heterologous dengue serotype. Viral particles encountered as free viruses and complexes (V+4· IgM-DENV+ IgG-DENV). Free IgM is shown in violet and free IgG in green. Antibody–virus complexes, IgM-DENV and IgG-DENV, are shown in blue and orange, respectively. The initial values are set the same as previous simulations.

**Figure 10 biology-10-00941-f010:**
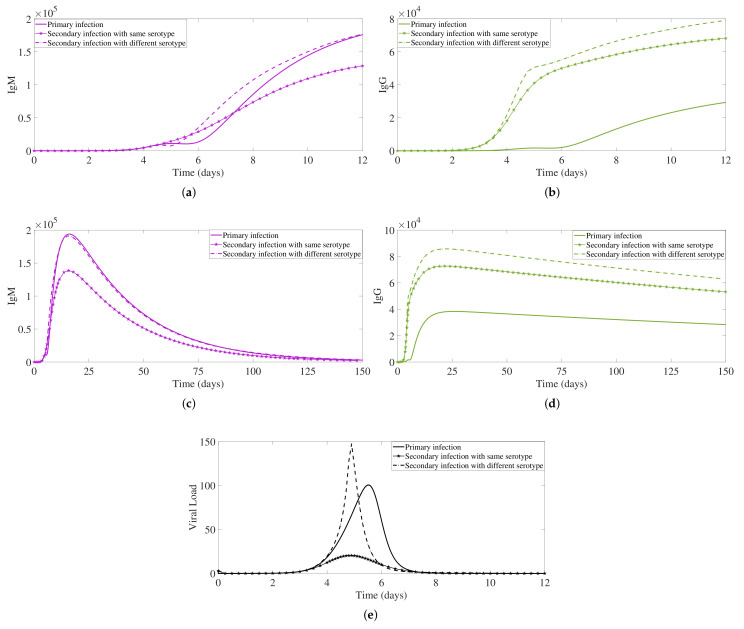
Antibody responses and free viral load comparison. For a primary dengue infection (full line), secondary infection with a homologous dengue serotype (line with pentagram marker), and secondary infection with heterologous dengue serotype (dashed line), we show the dynamics for IgM antibody type (in violet) and IgG antibody type (in green). In (**a**,**b**), we plot the antibodies dynamics over a 12-day period while in (**c**,**d**) over a 150-day period. Free viral load dynamics are shown in (**e**).

**Table 1 biology-10-00941-t001:** The biological meaning of the parameters and parameter values used for the numerical simulation.

Dengue Modeling Framework Parameters			
**Parameters**	**Parameter Values**	**Biological Meaning**	**References**
$\pi_{S}$	600	constant target cell production (monocytes/dendritic cells) per day	[45,52]
$\pi_{M}$	300	constant target cell production (macrophages) per day	[45,52]
$\mu_{S}$	1/30	lifespan of susceptible target cells in days	[48]
$\mu_{i}$	2	lifespan of infected cells (monocytes/dendritic cells) per day	modeled
$\mu_{P}$	$0.1\cdot \mu_{1}$	lifespan of presenting cells per day	modeled
$a=a_{m}$	0.02	infection rate of susceptible target cells per viral particle per day	modeled
$b=b_{m}$	15·a	removal rate of viral particles during the infection of target cells	modeled
κ	50	viral replication factor	[53]
$\alpha_{M}$	10	reproduction rate of IgM antibody per day	modeled
$\alpha_{G}$	1.5	reproduction rate of IgG antibody per day	modeled
$\alpha_{G_{s}ec}$	2000	activation rate of pre-existing IgG antibody per day	modeled
$\gamma_{M}=\gamma_{G}$	0.06	antibodies binding rate per day	modeled
$d_{M}$	$4\cdot \gamma_{M}$	binding rate of free virus with IgM antibody per day	modeled
$d_{G}$	$\gamma_{G}$	binding rate of free virus with IgG antibody per day	modeled
$\mu_{M}$	0.03	decay rate of IgM per day	[33]
$\mu_{G}$	1/365	decay rate of IgG per day	[33]
$\mu_{CM}=\mu_{CG}$	1	decay rate of antibody-virus complexes per day	modeled
$S(t_{0})$	$\pi_{S}/\mu_{S}$	initial value for target cells (monocytes/dendritic cells)	[48]
$S_{m}\left(t_{0}\right)$	$\pi_{M}/\mu_{S}$	initial value for target cells (macrophages)	[48]
$V(t_{0})$	3	initial value for free viral particles upon infection (mosquito bite)	modeled

## Data Availability

Data sharing not applicable.

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
