# Peer review of "Modeling Dengue Immune Responses Mediated by Antibodies: A Qualitative Study"

_biology, 2021, doi:10.3390/biology10090941_

Round 1

Reviewer 1 Report

This manuscript gives a significant contribution to science. The authors explained the dengue virus path in human organisms in a very simple and understandable way.  The authors wrote the manuscript very carefully and therefore no mistakes were found in the text. The English of the manuscript is very good as well. There are only a few suggestions for correction. 

The manuscript is only missing an explanation what data did authors use to make a model. This is given in a conclusion in only two sentences. This explanation should be given at the beginning of the manuscript. The authors should give a more detailed explanation. 

Reviewer 2 Report

The manuscript ‘Modeling Dengue Immune Responses Mediated by Antibodies: a Qualitative Study’ presents mathematical models for the immune response for primary and secondary infections of Dengue. The authors try to explain the differences of the immune response leading to different infection outcomes depending on if it is a primary infection or a secondary infection, with homologous or heterologous serotype.  The model represents the immunological response comprehensive way, simpler than others [42, 47] but that still includes the antibody-dependent enhancement process allows to explain the observations on different infection outcomes. Still, some of the modeling choices require further justification.

Major comments

I should say that I am not a specialist in immunology, so my comments are only on the modeling aspect. Some of the immunological processes are not very clearly explained so different formulations could explain observations.

  • The authors claim that for the secondary dengue infection with a homologous serotype the difference lies in the order of antibody production triggered by antigen presentation, shown in steps 4 to 7 and that immunological response initiates with IgG antibody type increasing quicker than the IgM type and reaching much higher levels than in primary infection (lines 188-194). So the authors include an extra term \alpha_G_sec V representing the pre-existing IgG antibodies that were produced during the primary dengue infection (line 196-197). I cannot understand why the increase of antibodies IgG is proportional to the amount of V instead of proportional to the amount of presenting cells P, as in the first infection and as shown in process 4, Figure 6. If so, please justify. If it was proportional to P, this would be equivalent to use a higher value for \alpha_G (\alpha_G+\alpha_G_sec), which, I think, it is not appropriated either. From my point of view, this could be modeled by assuming that IgG at the beginning of the second infection (initial conditions G(0)) is not ‘zero’ but approximately the equilibrium value of the first infection (as in [47] for T cells). I suspect that this would explain a faster production of the IgG complexes and therefore a faster clearance of the infection.

The same would be valid for the secondary dengue infection model with a heterologous serotype. In this case, this would explain a faster production of the IgG complexes and, since they are unable to eliminate the virus, it would enhance the free viral load release, explaining the haemorrhagic symptoms.

I would like to challenge the authors either to give a more clearly immunologic justification of this process and make the model and schematic representation coherent or to try other formulations.

  • For the secondary dengue infection model, it could more accurate to distinguish between IgG-DENV complexes in steps 6 and 9. The authors assume that both processes occur at the same time. I presume that those IgG already produced in the first infection are responsible for step 6 and those newly produced are responsible for step 9. It is not clear. Note that it could even occur some competition between them, mediated by the time occurred from the first infection and the amount of IgG already produced. I do not know if this would have any sense immunologically.

Please include more references to justify schemes presented in Figures 6 and 8 and further explain these processes.

Specific comments

  • Line 69: Include also citation [47] and Clapham et al.,2016, Plos Cop Biol
  • Figure 1: In the first part, in process 2 it is indicated \muk_i instead of \mu_i k – release of free virus
  • Line 103: use process 1 instead of step 1 to be consistent with others
  • Line 110-111: Should read ‘Presenting cells are assumed to trigger, via antigen presentation, the production of antibodies IgM (M) and IgG (G) with rates \alpha_gM and \alpha_gG respectively’ and not ‘…rates \gamma_gM and \gamma_gG respectively’
  • Table 1. In the reference, column indicate only references. Using ‘estimated’ it seems that some estimation procedure was conducted, which is not the case
  • System of equations (2): in the 3rd equation it is missing the term –d_G VG
  • Line 157: should read ‘including natural removal for IgM, \mu_M M, and IgG, \mu_G G.’ or ‘including a natural removal rate for IgM, \mu_M, and IgG, \mu_G.’
  • Figure 6 is not coherent with model 4.
  • Line 203: ‘\alpha_G_sec V=2000’ is not coherent with table 1.
  • Not all initial conditions for the different models are stated, please complete.

Round 2

Reviewer 2 Report

Thank you for your responses to my concerns and doubts. I fill enlighten by your responses and I think that the final text is now more complete and clear to the reader.

Minor comments:

  • In the abstract you use DENV-1, in the text DENV1 (line 37) and DENV-serotype throughout the text. Please uniform notation.
  • Line 222, line 244 and line 287 – include references used in the introduction for the respective immunological processes.
